# ROTATION-INVARIANT CLUSTERING OF NEURONAL RESPONSES IN PRIMARY VISUAL CORTEX

**Ivan Ustyuzhaninov,[1-3] Santiago A. Cadena,[1-3] Emmanouil Froudarakis,[4,5] Paul G. Fahey,[4,5]
Edgar Y. Walker,[4,5] Erick Cobos,[4,5] Jacob Reimer,[4,5] Fabian H. Sinz,[4,5]
Andreas S. Tolias,[1,4-6] Matthias Bethge,[1-3,5,†] Alexander S. Ecker[1-3,5,†,‡,*]**

[1] Centre for Integrative Neuroscience, University of Tübingen, Germany
[2] Bernstein Center for Computational Neuroscience, University of Tübingen, Germany
[3] Institute for Theoretical Physics, University of Tübingen, Germany
[4] Department of Neuroscience, Baylor College of Medicine, Houston, TX, USA
[5] Center for Neuroscience and Artificial Intelligence, BCM, Houston, TX, USA
[6] Department of Electrical and Computer Engineering, Rice University, Houston, TX, USA

† *Authors contributed equally*
‡ *Present address: Department of Computer Science, University of Göttingen, Germany*

\* ecker@cs.uni-goettingen.de

## ABSTRACT

Similar to a convolutional neural network (CNN), the mammalian retina encodes visual information into several dozen nonlinear feature maps, each formed by one ganglion cell type that tiles the visual space in an approximately shift-equivariant manner. Whether such organization into distinct cell types is maintained at the level of cortical image processing is an open question. Predictive models building upon convolutional features have been shown to provide state-of-the-art performance, and have recently been extended to include rotation equivariance in order to account for the orientation selectivity of V1 neurons. However, generally no direct correspondence between CNN feature maps and groups of individual neurons emerges in these models, thus rendering it an open question whether V1 neurons form distinct functional clusters. Here we build upon the rotation-equivariant representation of a CNN-based V1 model and propose a methodology for clustering the representations of neurons in this model to find functional cell types independent of preferred orientations of the neurons. We apply this method to a dataset of 6000 neurons and visualize the preferred stimuli of the resulting clusters. Our results highlight the range of non-linear computations in mouse V1.

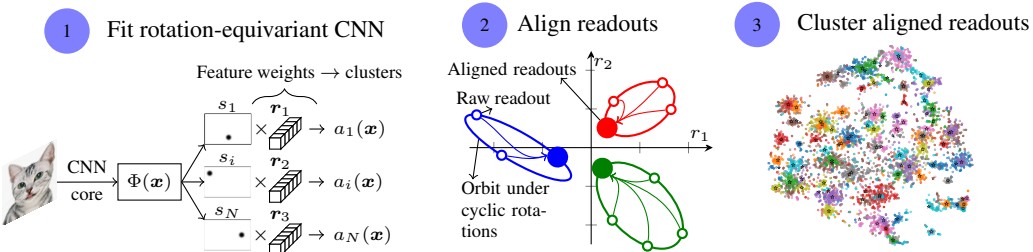

Figure 1: An overview of our approach. ① Fit rotation-equivariant CNN to predict neural responses and use readout vectors $r_i$ as proxies for neural computations. ② Align readouts to account for different preferred orientations. ③ Cluster the aligned readouts.

# 1 INTRODUCTION

A compact description of the nonlinear computations in primary visual cortex (V1) is still elusive. Like in the retina (Baden et al., 2016; Sanes & Masland, 2015), such understanding could come from a functional classification of neurons. However, it is currently unknown if excitatory neurons in V1 are organized into functionally distinct cell types.

It has recently been proposed that predictive models of neural responses based on convolutional neural networks could help answer this question (Klindt et al., 2017; Ecker et al., 2019). These models are based on a simple principle (Fig. 1-①): learn a *core* (e.g. a convolutional network) that is shared among all neurons and provides nonlinear features $\Phi(x)$, which are turned into predictions of neural responses by a linear *readout* for each neuron (Antolík et al., 2016). Models based on this basic architecture exploit aspects of our current understanding of V1 processing. First, convolutional weight sharing allows us to characterize neurons performing the same computation but with differently located receptive fields by the same feature map (Klindt et al., 2017; McIntosh et al., 2016; Kindel et al., 2019; Cadena et al., 2019). Second, V1 neurons can extract local oriented features such as edges at different orientations, and most low-level image features can appear at arbitrary orientations. Therefore, Ecker et al. (2019) proposed a rotation-equivariant convolutional neural network model of V1 that extends the convolutional weight sharing to the orientations domain.

The basic idea of previous work (Klindt et al., 2017; Ecker et al., 2019) is that each convolutional feature map could correspond to one cell type. While this idea is conceptually appealing, it hinges on the assumption that V1 neurons are described well by individual units in the shared feature space. However, existing models do not tend to converge to such solutions. Instead, V1 neurons are better described by linearly combining units from the same spatial location in multiple different feature maps (Ecker et al., 2019). Whether or not there are distinct functional cell types in V1 is therefore still an open question.

Here, we address this question by introducing a clustering method on rotation-equivariant spaces. We treat the feature weights (Fig. 1-①) that map convolutional features to neural responses as an approximate low-dimensional vector representation of this neuron's input-output function. We then split neurons into functional types using a two-stage procedure: first, because these feature weights have a rotation-equivariant structure, we find an alignment that rotates them into a canonical orientation (Fig. 1-②); in a second step, we cluster them using standard approaches such as k-means or Gaussian mixture models (Fig. 1-③). We apply our method to the published model and data of Ecker et al. (2019) that contains recordings of around 6000 neurons in mouse V1 under stimulation with natural images. Our results suggest that V1 neurons might indeed be organized into functional clusters. The dataset is best described by a GMM with around 100 clusters, which are to some extent redundant but can be grouped into a smaller number of 10–20 groups. We analyse the resulting clusters via their maximally exciting inputs (MEIs) (Walker et al., 2018) to show that many of these functional clusters do indeed correspond to distinct computations.

# 2 RELATED WORK

**Unsupervised functional clustering via system identification**    As outlined in the introduction, our work builds directly upon the methods developed by Klindt et al. (2017) and Ecker et al. (2019). While these works view the feature weights as indicators assigning each neuron to its 'cell type' (feature map), we here take a different view on the same model: rather than focusing on the convolutional features and viewing them as cell types, we treat the feature weights as a low-dimensional representation of the input-output function of each neuron and perform clustering in this space. This view on the problem has the advantage that there is no one-to-one correspondence between the number of feature maps and the number of cell types and we disentangle model fitting from its interpretation. On the other hand, our approach comes with an addition complexity: because the feature weights obey rotational equivariance and we would like our clustering to be invariant to rotations, we require a clustering algorithm that is invariant with respect to a class of (linear) transformations.

**Invariant clustering**    A number of authors have developed clustering methods that are invariant to linear (Tarpey, 2007), affine (Brubaker & Vempala, 2008) or image transformations by rotations,

scalings and translations (Frey & Jojic, 2002). Ju et al. (2019) cluster natural images by using a CNN to represent the space of invariant transformations rather than specifying it explicitly.

**Alignments**   Instead of using custom clustering algorithms that are invariant under certain transformations, we take a simpler approach: we first transform our features such that they are maximally aligned using the class of transformations the clustering should be invariant to. This approach has been used in other contexts before, usually by minimizing the distances between the transformed observations. Examples include alignment of shapes in $\mathbb{R}^d$ using rigid motions (Gower, 1975; Dryden & Mardia, 1998), alignment of temporal signals by finding monotonic input warps (Zhou & De la Torre, 2012), or alignment of manifolds with the distance between the observations being defined according to the metric on the manifold (Wang & Mahadevan, 2008; Cui et al., 2014). There is also work on alignment objectives beyond minimizing distances between transformed observations, examples of which include objectives based on generative models of observations (Kurtek et al., 2011; Duncker & Sahani, 2018) or probabilistic ones which are particularly suited for alignment with multiple groups of underlying observations (Kazlauskaite et al., 2019).

## 3   ROTATION-EQUIVARIANT CLUSTERING

Our goal is to cluster neurons in the dataset into groups performing similar computations. To do so, we use their low-dimensional representations obtained from the published rotation-equivariant CNN of Ecker et al. (2019), which predicts neural activity as a function of an external image stimulus. We briefly review this model before describing our approach to rotation-invariant clustering.

**CNN model architecture**   The model consists of two parts (Fig. 1-①):

1. A convolutional *core* that is shared by all neurons and computes feature representations $\Phi(\boldsymbol{x}) \in \mathbb{R}^{W \times H \times K}$, where $\boldsymbol{x}$ is the input image, $W \times H$ is the spatial dimensionality and $K$ is the number of feature maps.
2. A separate linear *readout* $\boldsymbol{w}_n = \boldsymbol{s}_n \otimes \boldsymbol{r}_n \in \mathbb{R}^{W \times H \times K}$ for each neuron $n = 1, \ldots, N$, factorized into a spatial mask $\boldsymbol{s}_n$ and a vector of feature weights $\boldsymbol{r}_n$.

The predicted activity of a neuron $n$ for image $\boldsymbol{x}$ is

$$a_n(\boldsymbol{x}) = f(\boldsymbol{w}_n \cdot \Phi(\boldsymbol{x})) = f(\boldsymbol{r}_n \cdot \boldsymbol{s}_n \cdot \Phi(\boldsymbol{x})) \in \mathbb{R} \tag{1}$$

where $f(\cdot)$ is a non-linear activation function. Such a CNN therefore provides $K$-dimensional feature weights $\boldsymbol{r}_n$ characterising linear combinations of spatially weighted image features $\boldsymbol{s}_n \cdot \Phi(\boldsymbol{x})$ that are predictive of neural activity. We treat these feature weights as finite-dimensional proxies of actual computations implemented by the neurons. Because masks $\boldsymbol{s}_n$ (another component of readouts $\boldsymbol{w}_n$ defined above) are irrelevant for our analysis, we will often refer to $\boldsymbol{r}_n$ simply as readout vectors. We will be referring to the matrix having $\boldsymbol{r}_i$ as its rows as the readout matrix $\boldsymbol{R} \in \mathbb{R}^{N \times K}$.

**Rotation-equivariant core**   Feature representations $\Phi(\boldsymbol{x})$ are rotation-equivariant, meaning that weight sharing is not only applied across space but also across rotations: for each convolutional filter there exist $O$ rotated copies, each rotated by $2\pi/O$. Feature vectors $\phi_{ij}(\boldsymbol{x})$ at position $(i, j)$ therefore consist of $F$ different features, each computed in $O$ linearly-spaced orientations (such that $F \times O = K$). We can think of $\phi_{ij}(\boldsymbol{x})$ as being reshaped into an array of size $F \times O$. Having computed $\phi_{ij}(\boldsymbol{x})$, we can compute $\phi_{ij}(\boldsymbol{x}')$ with $\boldsymbol{x}'$ being a stimulus $\boldsymbol{x}$ rotated around $(i, j)$ by $2\pi/O$ by cyclically shifting the last axis of $\phi_{ij}(\boldsymbol{x})$ by one step (this mechanism is illustrated in Fig. 2).

**Rotation-equivalent computations**   The linear readout adheres to the same rotation-equivariant structure as the core. As our goal is to cluster the neurons by the patterns of features they pool while being invariant to orientation, we need to account for the set of weight transformations that correspond to a rotation of the stimulus when clustering neurons. We illustrate this issue with a small toy example consisting of six neurons that fall into two cell types (Fig. 2B). Within each cell type (columns in Fig. 2B), the individual neurons differ only in their orientation. More formally, we define the computations performed by two neurons $n_i$ and $n_j$ to be *rotation-equivalent* if there exists a rotation $\psi_{ij}$ such that for any input stimulus $\boldsymbol{x}$ we have $a_{n_i}(\boldsymbol{x}) = a_{n_j}(\psi_{ij}(\boldsymbol{x}))$. We will refer to such neurons as rotation-equivalent as well.

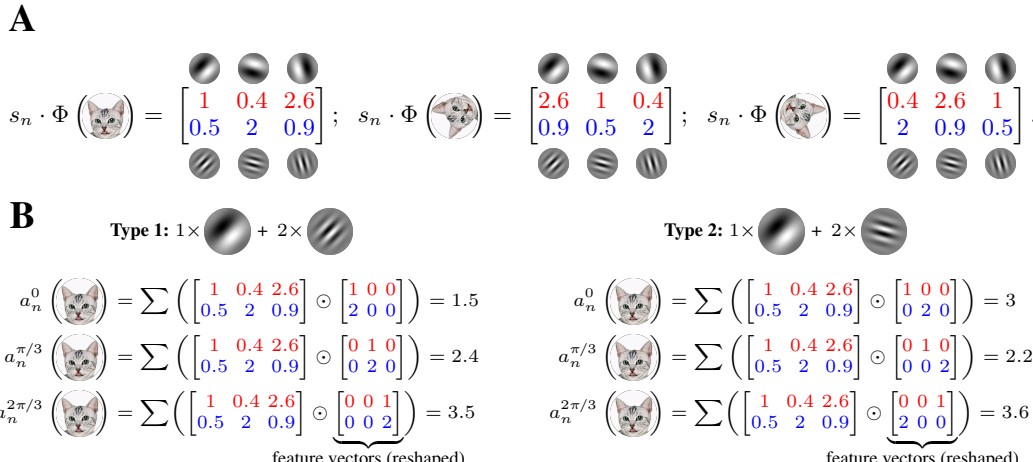

Figure 2: Toy example illustrating the computations in a rotation-equivariant CNN with two features (red and blue; cartoon feature representations are shown on top of corresponding values of $\Phi(\boldsymbol{x})$) in three orientations $(0, \pi/3, 2\pi/3)$. **A:** Output of the rotation-equivariant CNN for an input image rotated by $\pi/3$ (base orientation) can be computed by a cyclic shift. **B:** Example of two distinct types of neurons (columns) in three orientations (rows). Computations for both types consist of linear combinations of the two features computed by the CNN with the same weights, but in different relative orientations. Readouts of neurons of the same type in different orientations are cyclic shifts of each other, since they produce the same outputs on correspondingly rotated inputs.

**Readouts of rotation-equivalent neurons**  Directly clustering the readout matrix $\boldsymbol{R}$ does not respect the rotation equivalence, because readout vectors of neurons implementing rotation-equivalent computations are not identical (Fig. 2). To address that, we first modify $\boldsymbol{R}$ to obtain a matrix $\tilde{\boldsymbol{R}}$ with the rows corresponding to rotation-equivalent neurons aligned to a canonical form, and then cluster $\tilde{\boldsymbol{R}}$ to obtain functional cell types. Rotating an input $\boldsymbol{x}$ by a multiple of $2\pi/O$ corresponds to cyclically shifting $\Phi(\boldsymbol{x})$, so the readout vectors of two rotation-equivalent neurons whose orientation difference $\psi_{ij}$ is a multiple of $2\pi/O$ are also cyclic shifts of each other (Fig. 2). For arbitrary rotations that are not necessarily a multiple of $2\pi/O$, we assume the readout $\boldsymbol{r}_{n_j}$ of neuron $n_j$ to be a linear interpolation of cyclic shifts of $\boldsymbol{r}_{n_i}$ corresponding to the two nearest rotations which are multiples of $2\pi/O$. Formally, we define a *cyclic rotation matrix by an angle* $\alpha \in [0, 2\pi)$ as follows (matrix has shape $O \times O$; column indices are shown above the matrix):

$$
S_\alpha = \begin{matrix} 1 & & i & i+1 & i+2 & i+3 & & O \\ \begin{pmatrix} 0 & \cdots & 0 & 1-\gamma & \gamma & 0 & \cdots & 0 \\ 0 & \cdots & 0 & 0 & 1-\gamma & \gamma & \cdots & 0 \\ & \vdots & & & & & \vdots & \\ 0 & \cdots & 1-\gamma & \gamma & 0 & 0 & \cdots & 0 \end{pmatrix} \end{matrix}, \qquad \begin{aligned} i &= \left\lfloor \frac{\alpha O}{2\pi} \right\rfloor \mod O, \\ \gamma &= \frac{\alpha O}{2\pi} - i. \end{aligned} \tag{2}
$$

Given a readout vector $\boldsymbol{r}_n \in \mathbb{R}^K$, we can think of it as a matrix $\boldsymbol{r}_n \in \mathbb{R}^{O \times F}$ with columns corresponding to readout coefficients for different orientations of a single feature. The *cyclic rotation of a readout* $\boldsymbol{r}_n$ *by an angle* $\alpha \in [0, 2\pi)$ can be expressed as a matrix multiplication $\boldsymbol{r}_n(\alpha) = S_\alpha \boldsymbol{r}_n$. Note, by writing $\boldsymbol{r}_n(\alpha)$ as a function of $\alpha$ we refer to cyclically rotated (transformed) readouts, while $\boldsymbol{r}_n$ are the fixed ones coming from a pre-trained CNN and $\boldsymbol{r}_n(0) = \boldsymbol{r}_n$.

For two rotation-equivalent neurons $n_i$ and $n_j$, the readout vector $\boldsymbol{r}_{n_j}$ can be computed as a cyclic rotation of $\boldsymbol{r}_{n_i}$ by $\alpha$, which is the rotation angle of $\psi_{ij}$. If $\alpha$ is a multiple of $2\pi/O$; otherwise it is only an approximation which becomes increasingly accurate as $O$ increases.

**Aligning the readouts**  Assuming V1 neurons form discrete functional cell types, all neurons in the dataset (and hence the readouts characterising them) can be partitioned into non-overlapping classes w.r.t. the rotation equivalence relation we introduced above. Choosing one representative of each class and replacing the rows of $\boldsymbol{R}$ with their class representatives, we can obtain $\tilde{\boldsymbol{R}}$ from $\boldsymbol{R}$. Next, we discuss an algorithm for finding such representatives of each class.

We claim that by minimising the sum of pairwise distances between the cyclically rotated readouts

$$\{\alpha_i^*\} = \arg\min_{\{\alpha_i\}} \sum_{i=1}^{N} \sum_{j=i+1}^{N} ||\boldsymbol{r}_i(\alpha_i) - \boldsymbol{r}_j(\alpha_j)||, \tag{3}$$

we can transform each readout into a representative of a corresponding equivalence class (same for all readouts of a class), i.e. $\boldsymbol{r}_i(\alpha_i^*) = \boldsymbol{r}_j(\alpha_j^*)$ if neurons $i$ and $j$ are rotation-equivalent. This is indeed the case because the readouts of the same equivalence class lie on the orbit obtained by cyclically rotating any representative of that class. Such angles $\{\alpha_i^*\}$ that neurons of the same class end up on the same point on the orbit (i.e. aligned to the same class representative) clearly minimise Eq. (3), and since different orbits do not intersect (they are different classes of equivalence), readouts of different equivalence classes cannot end up at the same point. Note that the resulting class representatives are not arbitrary, but those with the smallest sum of distances between each other in the Euclidean space. This mechanism is illustrated in Fig. 1-②.

**Clustering aligned readouts** Having obtained $\tilde{\boldsymbol{R}}$ with rows $\boldsymbol{r}_i(\alpha_i^*)$, we can cluster the rows of this matrix using any standard clustering method (e.g. K-Means, GMM, etc.) to obtain groups of neurons (cell types) performing similar computations independent of their preferred orientations.

**Continuous relaxation of cyclic rotations** The rotation-invariant clustering described above is based on solving the optimisation problem in Eq. (3). To do so, we would typically use a gradient-based optimisation, which is prone to local minima because of the way we define cyclic rotations in Eq. (2). According to that definition, a rotated readout is a linear combination of two nearest base rotations, or rather a linear combination of all such rotations with only two coefficients being non-zero. That means that gradients of all but two coefficients w.r.t. the angle $\alpha$ are zero, and the optimisation would converge to a local minimum corresponding to the best linear combination of the two base rotations initialised with non-zero coefficients (Fig. 3).

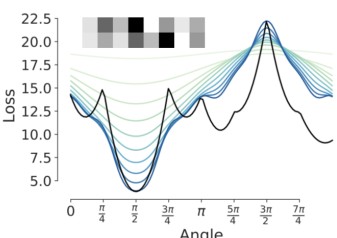

Figure 3: Distance between two vectors (top left corner) with first one fixed and second cyclically shifted by an angle on the x-axis. Continuous relaxation (shades of blue) of linearly interpolated (black) cyclic shifts smooths gradients and helps overcome local minima.

To address this issue, we propose to approximate Eq. (2), such that $\boldsymbol{r}_{n_i}(\alpha)$ is a linear combination of all base rotations with non-zero coefficients, with the coefficients for the two nearest base rotations being the largest. Specifically we compute the coefficients by sampling the von Mises density at fixed orientations to ensure cyclic boundary conditions and define $\tilde{\boldsymbol{r}}_{n_i}(\alpha)$, a continuous relaxation of $\boldsymbol{r}_{n_i}(\alpha)$, as $\tilde{\boldsymbol{r}}_{n_i}(\alpha) = \tilde{S}_\alpha \boldsymbol{r}_{n_i}$ where

$$\tilde{S}_\alpha = \begin{pmatrix} \gamma_1 & \gamma_2 & \cdots & \gamma_O \\ \gamma_O & \gamma_1 & \cdots & \gamma_{O-1} \\ \vdots & & & \vdots \\ \gamma_2 & \gamma_3 & \cdots & \gamma_1 \end{pmatrix} \quad \text{with} \quad \gamma_i = \frac{\exp(T\cos(\alpha - (i-1)\cdot 2\pi/O))}{\sum_{i=1}^{O} \exp(T\cos(\alpha - (i-1)\cdot 2\pi/O))}. \tag{4}$$

The parameter $T \geq 0$ controls the sparseness of coefficients $\{\gamma_i\}$. For small $T$, many of the coefficients are significantly greater than zero, allowing the optimiser to propagate the gradients and reduce the effect of initialisation. As $T$ increases, $\tilde{\boldsymbol{r}}_{n_i}(\alpha)$ becomes more similar to $\boldsymbol{r}_{n_i}(\alpha)$, and the rotations by multiples of $2\pi/O$ are recovered. In the limit, $\tilde{\boldsymbol{r}}_{n_i}(2\pi k/O) \to \boldsymbol{r}_{n_i}(2\pi k/O)$ as $T \to \infty$ (Fig. 3). Instead of fixing $T$, we learn it by optimising the regularised alignment objective with additional reconstruction loss preventing trivial solutions (e.g. all coordinates of $\tilde{\boldsymbol{r}}_i(\alpha_i)$ being the same for $T = 0$):

$$\{\alpha_i^*\} = \arg\min_{\{\alpha_i\}, T} \sum_{i=1}^{N} \sum_{j=i+1}^{N} ||\tilde{\boldsymbol{r}}_i(\alpha_i) - \tilde{\boldsymbol{r}}_j(\alpha_j)|| + \beta \sum_{i=1}^{N} ||\tilde{\boldsymbol{r}}_i(0) - \boldsymbol{r}_i||. \tag{5}$$

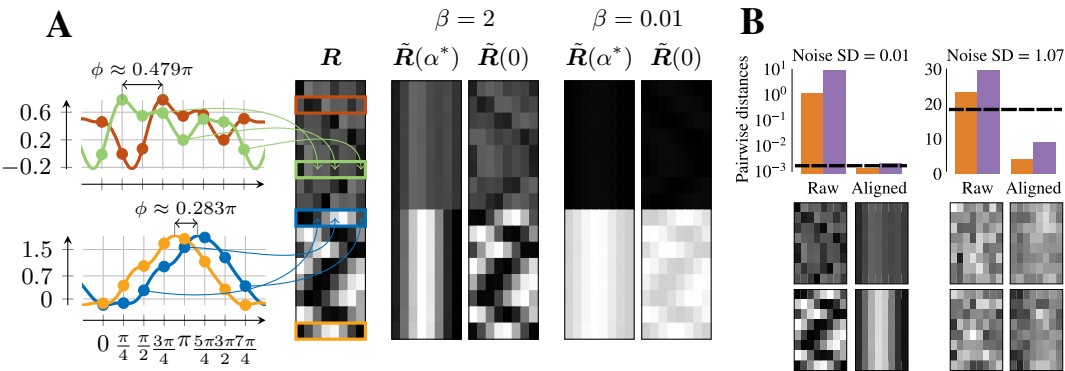

Figure 4: Synthetic data set: generation, alignment and dependence on noise. **A:** Panel $\boldsymbol{R}$ shows the unaligned synthetic data set as well as the corresponding shifted GP samples for each of the two groups of neurons (see details in the main text). Colored boxes in $\boldsymbol{R}$ correspond to the colors of corresponding GP samples. Panels $\tilde{\boldsymbol{R}}(\alpha^*)$ and $\tilde{\boldsymbol{R}}(0)$ show aligned readouts and readouts rotated by 0 using Eq. (4) respectively. $\tilde{\boldsymbol{R}}(0)$ should be similar to $\boldsymbol{R}$ for an adequate choice of $\beta$ (and consequently optimised value of $T$). **B:** Effect of observation noise. Means of pairwise distances for each of the two groups shown in matrix $\boldsymbol{R}$ for two levels of Gaussian noise added to the dataset. Black dashed line: expected pairwise distance due to noise only (i.e. for perfectly aligned data with added noise). Raw and aligned matrices for each of the two groups are shown below the bar plots.

## 4 EXPERIMENTS

**Synthetic dataset**    We generate a small toy dataset consisting of 16 hypothetical neurons (readouts) of two cell types to illustrate the proposed alignment method. Each readout consists of just one feature in eight base orientations (linearly spaced between 0 and $7\pi/4$) and is generated by one of the two underlying types of readouts cyclically shifted by a random angle $\phi \in [0, 2\pi)$. To generate such a dataset , we draw two independent noiseless functions from a Gaussian process (GP) with a periodic kernel (with period $2\pi$), then for each readout we randomly choose one of the two GP samples, shift it by an angle $\phi$ and evaluate the shifted function at the base orientations to obtain an 8-dimensional vector modelling the observed readout values. This process is illustrated in Fig. 4A.

**Neural data**    We use the same dataset as in Ecker et al. (2019), consisting of simultaneous recordings of responses of 6005 excitatory neurons in mouse primary visual cortex (layers 2/3 and 4).

**Model details**    We analyse a rotation-equivariant CNN consisting of a three-layer core with 16 features in 8 orientations in each layer (kernel sizes 13, 5, 5) and 128-dimensional readouts ($F = 16$, $O = 8$). We use the pre-trained model provided by Ecker et al. (2019). We align the readout matrix $\boldsymbol{R}$ by minimising Eq. (5) w.r.t. the rotation angles $\alpha_i$ and temperature $T$. We fit models for 20 log-spaced values of $\beta$ in $[0.001, 10]$, and choose for analysis the one with the smallest alignment loss (Eq. (3)) among the models with optimised temperature $T > 5$. We use Adam (Kingma & Ba, 2015) with early stopping and initial learning rate of 0.01 decreased three times.

**Clustering aligned readouts**    We use the Gaussian mixture model implemented in scikit-learn (Pedregosa et al., 2011) for clustering the aligned readouts $\tilde{\boldsymbol{R}}$. We use spherical covariances to reduce the number of optimised parameters. To obtain a quantitative estimate of the number of clusters in $\tilde{\boldsymbol{R}}$, we randomly split the dataset of 6005 neurons into training (4000 neurons) and test (2005 neurons) sets, fit GMMs with different numbers of clusters on the training set, and then evaluate the likelihood of the fitted model on the test set.

## 5 Results

### 5.1 Synthetic data set alignment

We start by demonstrating on a synthetic dataset (Sec. 4) that optimising Eq. (5) can successfully align the readouts (Fig. 4A), assuming $\beta$ has been chosen appropriately. Note that readouts have been shifted by arbitrary angles (not multiples of $\pi/4$ as demonstrated for readouts in colored boxes in Fig. 4A), and they are aligned precisely via interpolation Eq. (4). We can also see the effect of the parameter $\beta$, controlling the relative weight of the reconstruction term (i.e. similarity of readouts rotated by 0 degrees to the observations). Small values of $\beta$ incur a small cost for poor reconstructions resulting in small optimised values of $T$ and over-smoothed aligned readouts.

We next ask whether the alignment procedure still works in the presence of observational noise (Fig. 4B). For small to moderate noise levels (Fig. 4B, left), alignment reduces the pairwise distances to the level expected from the observation noise (shown at the top), confirming the visual impression (shown at the bottom) that alignment works well. For high noise levels (Fig. 4B, right), alignment breaks down as expected, and we observe overfitting to the noise patterns (shown by the pairwise distances after alignment dropping below the level expected from observation noise alone).

### 5.2 Mouse V1 dataset

**Clustering** We evaluate the GMM used to cluster $\tilde{R}$ for different numbers of clusters. The test likelihood starts to plateau at around 100 clusters (Fig. 5), so we use 100 clusters in the following.

**Visualization of clusters** To visualize the clustering result, we compute a two-dimensional t-SNE embedding (van der Maaten & Hinton, 2008) of the matrix of aligned readouts $\tilde{R}$, which is coloured according to the GMM clustering of $\tilde{R}$ with 100 clusters (Fig. 6). Note that we use the embedding only for visualization, but cluster 128D aligned readouts in $\tilde{R}$ directly. In addition to the embeddings, we also visualize the computations performed by some of the clusters by showing maximally exciting inputs (MEIs). We compute MEIs via activity maximisation (Erhan et al., 2009; Walker et al., 2018) and show the stimuli that maximally drive the 16 best-predicted neurons of each cluster. We ob-

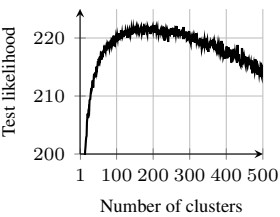

Figure 5: Test set likelihood of GMMs applied to $\tilde{R}$ as a function of the number of clusters.

serve that MEIs corresponding to neurons of the same cluster are generally consistent up to rotation and receptive field location, suggesting that the proposed clustering method captures the similarities in the neural computations while ignoring the nuisances as desired.

**Network learned redundant features** We noticed a number of clusters with similar MEIs (e.g. Block 9 and Block 13 in Fig. 6). There could be two reasons for this observation: (a) the neural computations corresponding to these clusters could be different in some other aspect, which we cannot tell by inspecting MEIs as they represent only the maximum of a complex function, or (b) the features learned by the CNN could be redundant, i.e. the hidden layers could learn to approximate the same function in multiple different ways. To answer this question, we compute a cluster confusion matrix (Fig. 7, left), which quantifies how similar the response predictions of different clusters are across images. The element $(p,q)$ corresponds to the correlation coefficient between the predicted responses on the entire training set of hypothetical neurons with cluster means for clusters $p$ and $q$ used as readouts, accounting for potential differences in canonical orientation across clusters. By greedily re-arranging clusters in the matrix into blocks based on their correlations, we show that the 100 clusters in the model can be grouped into a much smaller number of functionally distinct clusters. Using a correlation threshold of 0.5 in this re-arrangement procedure, we obtain an example arrangement into 17 blocks (Fig. 7). Thus, the network has learned an internal representation that allows constructing very similar functions in multiple ways, suggesting that further pruning the learned network before clustering could lead to a more compact feature space for V1.

Finally, to quantify how consistent the resulting 17 groups of clusters are, we compute an MEI confusion matrix (Fig. 7, right panel). Its $(i,j)$ element is the predicted activity of neuron $j$ for the MEI of neuron $i$, after accounting for orientation and receptive field location (i.e. $a_j(\boldsymbol{y}_i)$, where

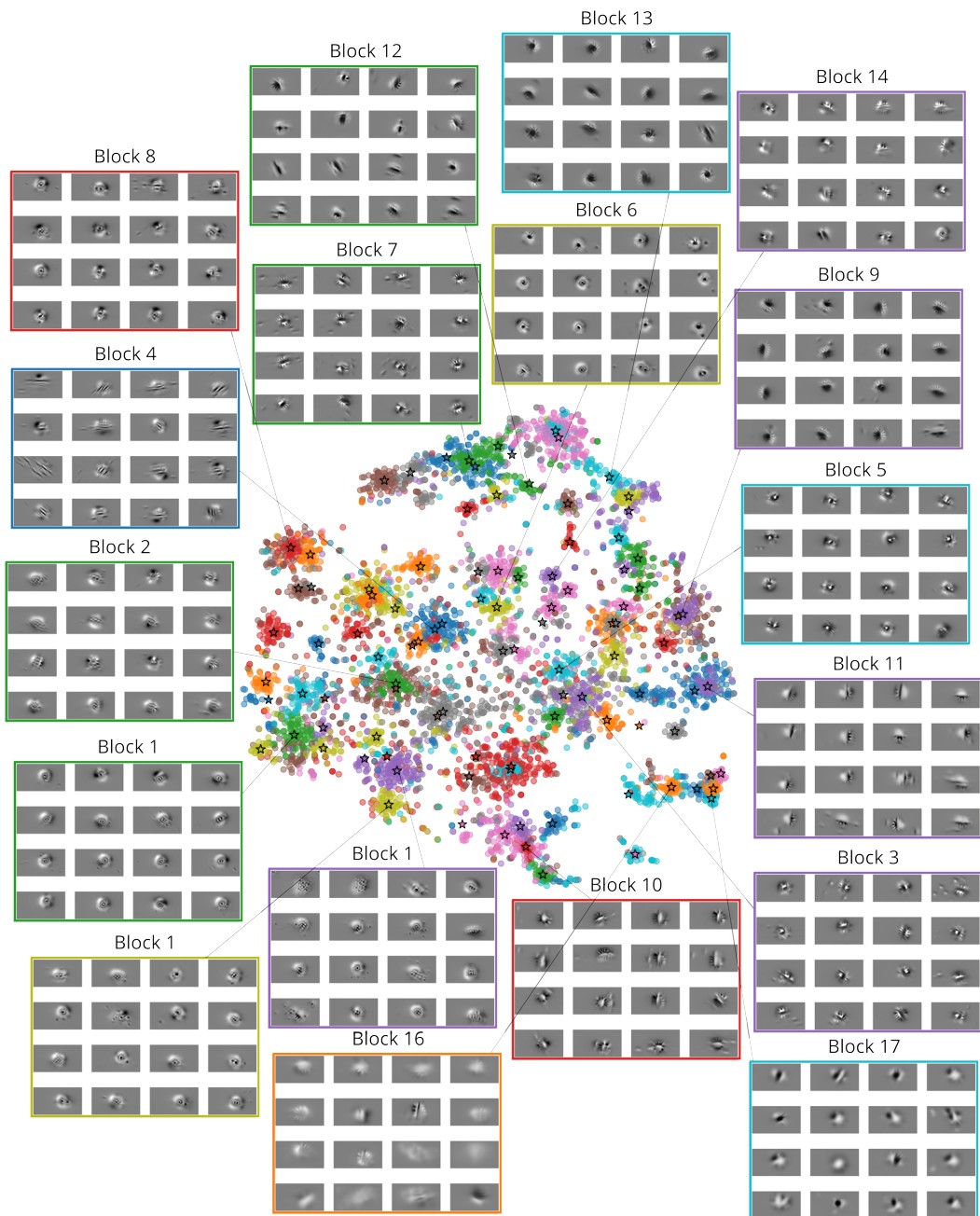

Figure 6: 2D t-SNE embedding of the aligned readouts $\tilde{R}$, colored according to the GMM clustering with 100 components. Black stars show the locations of cluster centers. For some of the clusters, the MEIs of 16 best predicted neurons of that cluster are shown. The titles in the MEI subfigures show which matrix block in Fig. 7 (left) that cluster belongs to.

$y_i$ is the MEI of neuron $i$ moved and rotated such that it optimally drives neuron $j$). We show this matrix using the same grouping as for the cluster confusion matrix above and restrict it to the 542 (out of 6005) best predicted neurons (with test set correlation $\geq 0.7$). Note that some of the blocks from the cluster confusion matrix do not appear here, indicating that those clusters include poorly predicted neurons (e.g. block 17). The MEI confusion matrix exhibits a block-diagonal structure, with most MEIs driving neurons within the same blocks most strongly, albeit with different degrees of within-block similarity.

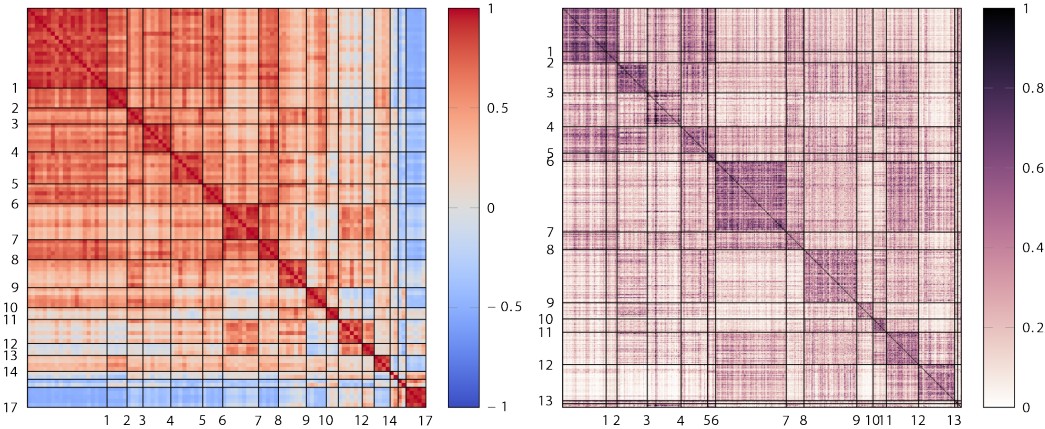

Figure 7: **Left:** Cluster confusion matrix ($100 \times 100$) for 100 clusters shown in Fig. 6. Rows and columns have been arranged into 17 groups (blocks). **Right:** MEI confusion matrix for well-predicted neurons (test correlation $\geq 0.7$) arranged into the same 17 blocks as on the left.

## 6 CONCLUSIONS AND FUTURE WORK

We have presented an approach to clustering neurons into putative functional cell types invariant to location and orientation of their receptive field. We find around 10–20 functional clusters, the boundaries of some of which are not very clear-cut. To systematically classify the V1 functional cell types, these proposals need to be subsequently examined based on a variety of biological criteria reflecting the different properties of the neurons and the prior knowledge about the experiment.

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

# A    RANDOM PERMUTATIONS OF FEATURES

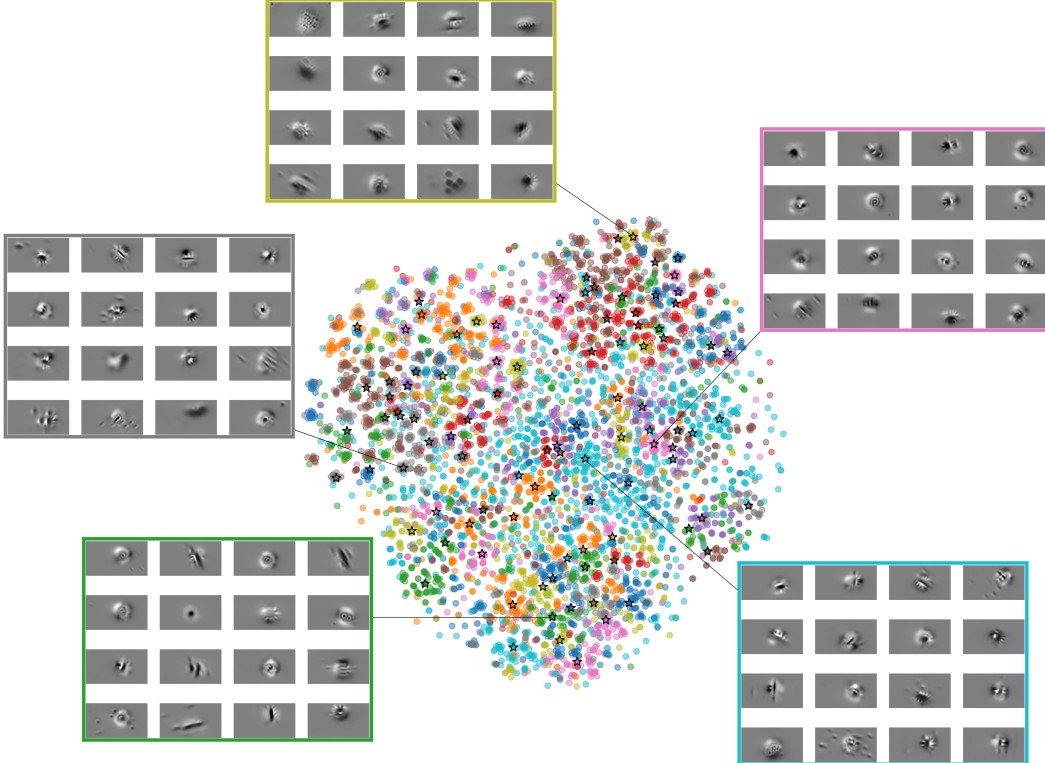

Figure A1: 2D t-SNE embedding of the aligned readouts $\tilde{R}$ with feature weights randomly permuted for each of the neurons. The colors correspond to the GMM clustering with 100 components. Black stars show the locations of cluster centers. For some of the clusters, the MEIs of 16 best predicted neurons of that cluster are shown.

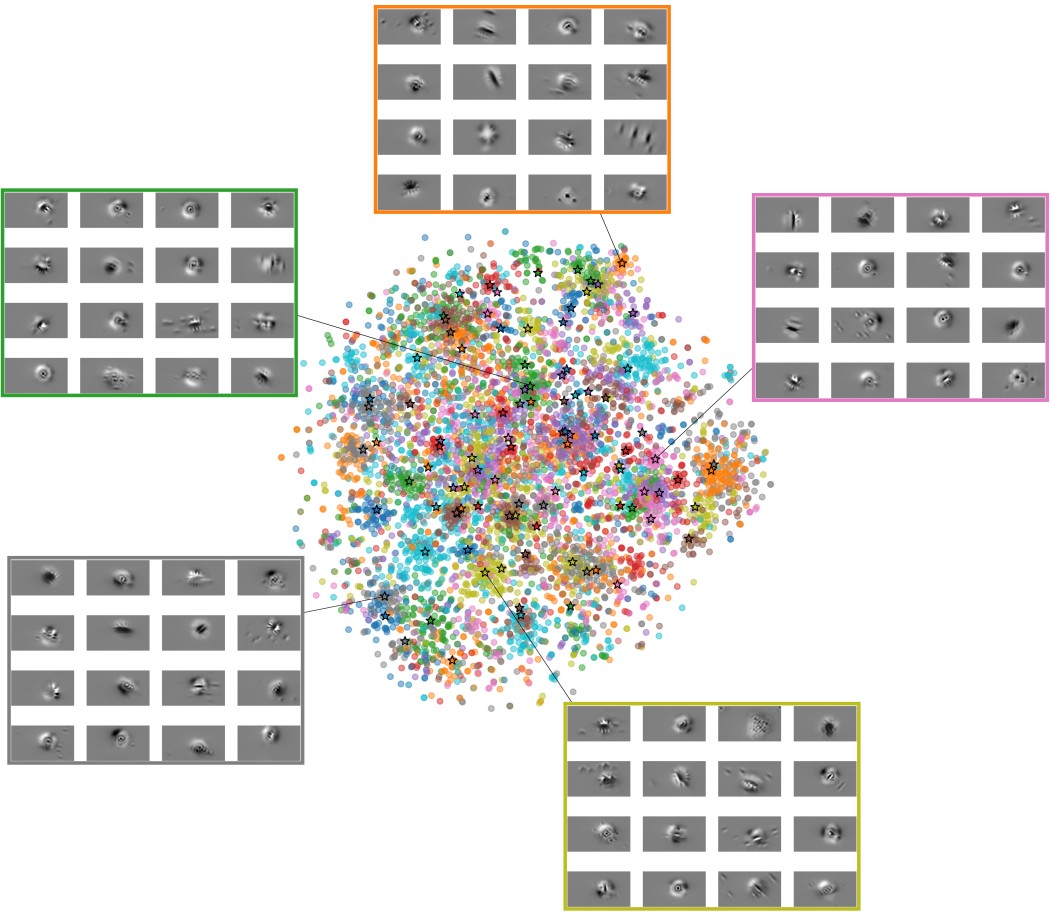

Figure A2: 2D t-SNE embedding of the aligned readouts $\tilde{R}$ with feature weights randomly permuted across the neurons. The colors correspond to the GMM clustering with 100 components. Black stars show the locations of cluster centers. For some of the clusters, the MEIs of 16 best predicted neurons of that cluster are shown.

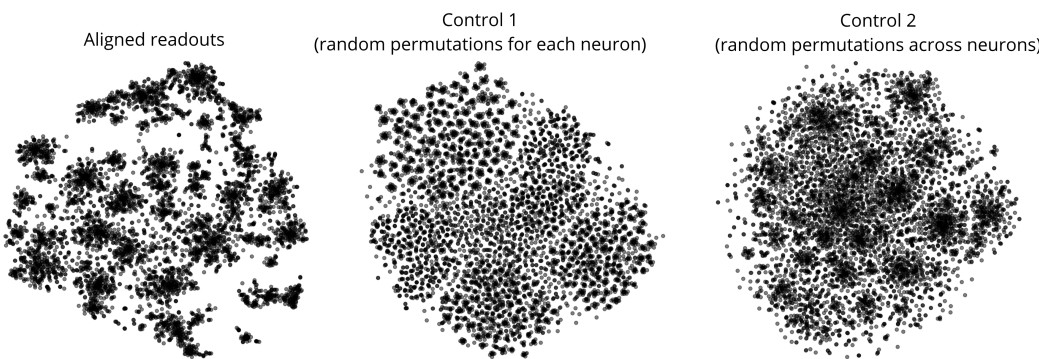

Figure A3: t-SNE embeddings for the aligned readouts (Fig. 6), and the controls with randomly permuted features for each neuron (Fig. A1) and across the neurons (Fig. A2).

## B  SYNTHETIC DATASET: DEPENDENCE ON NOISE

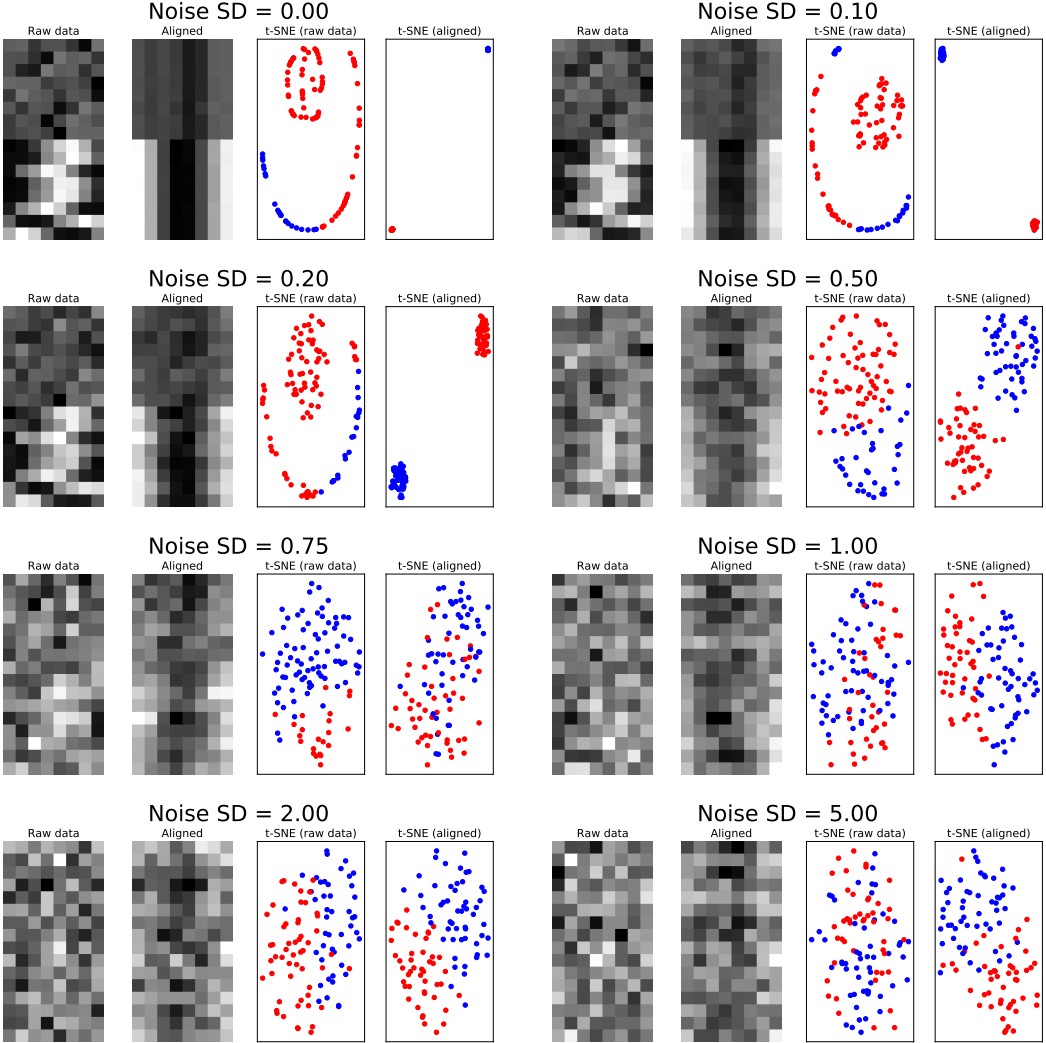

Figure B1: Alignment of a synthetic dataset of 100 observations generated using the procedure described in Sec. 4 for different amount of i.i.d. Gaussian noise added to the observations. The panels for each noise level show the 16 (out of 100) examples of the raw and aligned data as well the t-SNE embeddings of raw and aligned data coloured according to the GMM clustering with two components.

## C  MERGES AND SPLITS OF CLUSTER CONFUSION MATRIX BLOCKS

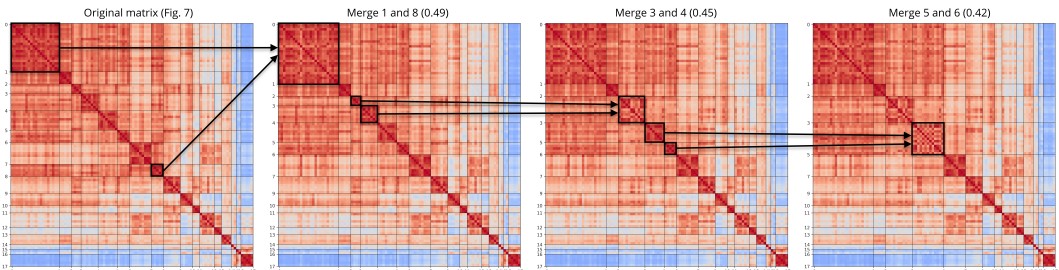

Figure C1: Sequential merges of the three pairs of blocks with the highest correlations in the cluster confusion matrix (Fig. 7, left). The merged blocks and the correlation values are shown in the titles of panels.

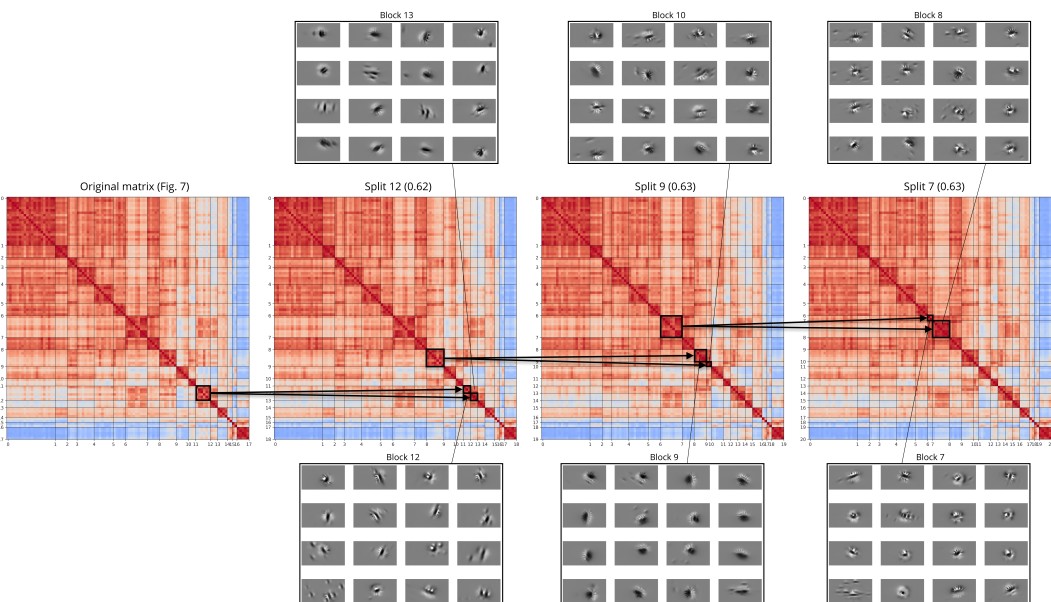

Figure C2: Sequential splits of the three pairs of blocks in the cluster confusion matrix (Fig. 7, left). The merged blocks, the correlation values, and the examples of MEIs of one of the GMM clusters in each of the splitted blocks are shown for each splitting step.

