# OpenReview forum: "Rotation-invariant clustering of neuronal responses in primary visual cortex"
_ICLR.cc/2020/Conference — Accept (Talk)_

### Official Review · AnonReviewer2 · 2019-10-23
**Official Blind Review #2**

**Rating:** 8

**Review:**

The authors present a rotation-invariant representation of a CNN modeling the V1 neurons and a pipeline to cluster these neurons to find cell types that are rotation-invariant. Experimental validation is performed on a 6K neuron dataset with promising results.
The paper is well postulated.

Below are comments about the work:

1. In Figure 2, what does 1 x feature + 2 x another_feature mean?
2. In Equation 3, why was the ‘square’ of error differences not used?
3. In the clustering approach, how is the number of mixtures set for the GMM? How stable is the model to different number of mixtures?
4. In Figure 6: are Blocks 5 and 13 the same clusters (since they are of the same color) or is it that the colourmap use did not have 100 colors?
5. In the ‘network learned redundant features’, Sentence 1: why do the authors say ‘similar MEIs’. The 16 neurons rendered in both blocks look different.
6. It will be informative to know how the number of clusters vary based on the correlation threshold used to collapse 100 clusters to a lower number. Are the clusters still functionally distinct for varying thresholds? Further why is MEI confusion matrix only shown for 13 groups?

**Experience Assessment:**

I have read many papers in this area.

**Review Assessment: Checking Correctness Of Derivations And Theory:**

I assessed the sensibility of the derivations and theory.

**Review Assessment: Checking Correctness Of Experiments:**

I assessed the sensibility of the experiments.

**Review Assessment: Thoroughness In Paper Reading:**

I read the paper at least twice and used my best judgement in assessing the paper.

---

> ### Author Response · Authors · 2019-11-09
> **A reply to Reviewer 2 (continuation)**
>
> >  6. It will be informative to know how the number of clusters vary based on the correlation threshold used to collapse 100 clusters to a lower number. Are the clusters still functionally distinct for varying thresholds? Further why is MEI confusion matrix only shown for 13 groups?
>
> The MEI confusion matrix is shown for a subset of the well predicted neurons (with test correlation >= 0.7), and some of the 17 groups shown in the cluster confusion matrix don’t contain such neurons, resulting in only 13 groups present in the MEI matrix. The motivation for that is that the MEIs are most informative for well-predicted neurons.
>
> The cluster confusion matrix shows all 100 GMM clusters, and the merging procedure with the 0.5 threshold is a heuristic to rearrange the rows and columns to highlight the block structure in the matrix. We tried different threshold values and chose the value of 0.5 by visual inspection of the resulting block structure in the matrix and the corresponding MEIs.
>
> To provide a better intuition of how the threshold value affects the matrix, we have updated the paper to include Figures C1 and C2. In Figure C1 we show the cluster confusion matrices resulting from starting the matrix in Figure 7 and merging the next three pairs of blocks with the highest correlations. We can see that such merges result in biggest blocks, which are not homogeneous (e.g. block 5 after the third merge clearly exhibits a checkerboard pattern suggesting it contains sufficiently different GMM clusters, which we want to avoid within the same block).
>
> In Figure C2 we show the sequential splits of the three blocks (which is equivalent to performing the initial merging procedure backwards starting from the matrix in Figure 7). The split blocks shown are not the last ones merged before achieving the correlation threshold of 0.5 (those are blocks 15 and 16, each of which contains only two GMM cluster, but rather the ones merged before them, which we think are more illustrative for this figure). The examples of the MEIs of the split blocks are also shown. The split blocks seem to be correlated with the MEIs being sufficiently similar (perhaps with the exception of blocks 9 and 10).
>
> Generally speaking, it is hard to find an exact threshold value which works best. However, grouping the cluster confusion matrix allows us to control the granularity of the functional blocks of neurons to highlight the range of computations implemented by the neurons and produce hypotheses of functional cell types which can be tested by further analysis based on other biological evidence (e.g. morphology or genetics).

---

> ### Author Response · Authors · 2019-11-09
> **A reply to Reviewer 2**
>
> Thank you for the review and the comments! Below we answer your questions.
>
>
> > 1. In Figure 2, what does 1 x feature + 2 x another_feature mean?
>
> Feature 1 (low frequency Gabor) and feature 2 (high frequency Gabor) are cartoon representations of the two features computed by the hypothetical rotation-equivariant CNN (CNN features are not simple Gabors of course, it is just a cartoon). The neurons are assumed to implement the linear combinations of the CNN outputs corresponding to those two features with coefficients 1 and 2. Our goal was to illustrate how different orientations of the neurons and the relative orientations of the CNN features influence the readout weights (i.e. linear combination weights) if the CNN is rotation-equivariant. We are sorry if the figure confused more than clarified this mechanism, and we are happy to answer any other questions about this mechanism.
>
>
> > 2. In Equation 3, why was the ‘square’ of error differences not used?
>
> We haven’t tried using squared difference. However, we don’t expect it to result in any substantial differences. Squared differences are more convenient to differentiate, but in times of automatic differentiation this benefit is not really relevant.
>
>
> > 3. In the clustering approach, how is the number of mixtures set for the GMM? How stable is the model to different number of mixtures?
>
> We evaluated the test likelihood on a held-out set of neurons for different numbers of clusters (Fig. 5), and chose 100 clusters as it is roughly where the likelihood curve started to saturate.
>
> We group neurons into groups performing similar computations by examining the MEIs and the confusion matrices (Fig. 7). We find that the number of such groups is much smaller than the number of GMM mixtures (17 vs. 100). This means that as long as the GMM uses sufficiently many clusters, they would be merged into larger groups during post-processing, therefore, the exact number of GMM mixtures is not important.
>
>
> > 4. In Figure 6: are Blocks 5 and 13 the same clusters (since they are of the same color) or is it that the colourmap use did not have 100 colors?
>
> The colormap doesn’t have 100 colors. We tried using a colormap with 100 different colors, however, many of them are very similar and hard to visually distinguish, so they don’t provide much additional information. Therefore we chose to use a colormap with clearly distinct colors and show the correspondence between the block and the cluster in the scatter plot by the color of the border in the MEIs subplot.
>
>
> > 5. In the ‘network learned redundant features’, Sentence 1: why do the authors say ‘similar MEIs’. The 16 neurons rendered in both blocks look different.
>
> The 16 MEIs shown in blocks 9 and 13 look visually similar to us. Could you elaborate on the differences you observed?

---

### Official Review · AnonReviewer3 · 2019-10-24
**Official Blind Review #3**

**Rating:** 8

**Review:**

In this study, the authors develop a method to cluster cells in primary visual cortex (V1) based on the cells' responses to natural images. The method consists in three steps:
- fit a rotation-equivariant convolutional neural network model to V1 cells (previously described in Ecker et al. 2019)
- align all cells by choosing the rotation for each cell that minimizes overall distance between cells in feature space, so that the clustering is mostly blind to the orientation of the filters.
- cluster the cells using a Gaussian mixture model (GMM).

Although I find this article mostly well-written and the topic important, I cannot recommend acceptance because (1) the study does not make a significant contribution to our understanding of V1, (2) the main innovation in ML presented (alignment method) is quite specific and will thus not likely be of interest for the general audience of ICLR:

(1) An important question in visual neuroscience is whether V1 cells form discrete functional clusters as opposed to a continuum. Another related question is whether these functional clusters correspond to distinct cell types characterized by specific wiring patterns, gene expression and/or morphology.
The analyses performed do not answer any of these two questions:
- the clustering model (GMM model) is not compared statistically to other models that would assume a continuous structure in the data (e.g. cells form a sparse continuous manifold in feature space). Although clusters do appear in the t-SNE visualization, this visualization does not provide statistical evidence that cells indeed form distinct clusters.
- The correspondence of the proposed clusters to cell types with specific wiring patterns, gene expression and/or morphology is not established. To establish this correspondence would require further experiments, as acknowledged by the authors: "To systematically classify the V1 functional cell types, these proposals need to be subsequently examined based on a variety of biological criteria reflecting the different properties of the neurons and the prior knowledge about the experiment".

(2) The alignment method, which consists in rotating the cells in feature space so that orientation is not a factor for subsequent clustering, is quite specific to the problem studied and likely not of interest for the general ICLR audience.


Additional feedback:

- Title: ROTATION-INVARIANT CLUSTERING OF FUNCTIONAL CELL TYPES IN PRIMARY VISUAL CORTEX
=> "functional cell types" is not adequate here, since the article does not establish the existence of functional cell types. Could be replaced with "cell responses".

- Abstract: We apply this method to a dataset of 6000 neurons and provide evidence that discrete functional cell types may exist in V1.
=> this sentence is misleading, since no evidence for functional clusters is provided.

- "Thus, the network has learned an internal representation that allows constructing very similar functions in multiple ways"
=> To avoid the caveat of redundant features, the authors could try to add a dimensionality bottleneck on feature space before readout.

- "Small values of β incur a small cost for poor reconstructions resulting in small optimised values of T and over-smoothed aligned readouts."
=> A simulated annealing procedure (progressive increase of T during learning) could potentially allow the use of larger β values here (i.e. less distortion of the filter).

- The alignment procedure could lead to the emergence of spurious structure in the clustering. It would be important to control for this potential artifact by running the procedure on an unstructured synthetic dataset.

- It is possible that the MEIs within clusters look more similar than they actually are, since the cells are fitted from the same common bank of features. It would be useful but maybe difficult to control for this.

- It would be interesting to test the clustering procedure on a shuffled version of the readout weights (shuffle across features and V1 cells), so as to keep sparsity but not any other structure. Does the t-SNE map look less clustered? Is the GMM fit qualitatively different?

- Fig1(2): add legend/caption. what are the ellipses?













**Experience Assessment:**

I have published one or two papers in this area.

**Review Assessment: Checking Correctness Of Derivations And Theory:**

I assessed the sensibility of the derivations and theory.

**Review Assessment: Checking Correctness Of Experiments:**

I carefully checked the experiments.

**Review Assessment: Thoroughness In Paper Reading:**

I read the paper thoroughly.

---

> ### Author Response · Authors · 2019-11-09
> **A reply to Reviewer 3**
>
> Thank you for the thorough review and the useful comments. Your main point of contention appears to be a perceived lack of evidence for functional cell types. While we agree that we do not provide undeniable proof, we do believe that the clusters revealed in Fig. 6 and the clear block-diagonal structure of the confusion matrices in Fig. 7 constitute important pieces of evidence that at least suggest that functional cell types may exist.
>
> Assessing whether discrete clusters exist in a high-dimensional space is a notoriously difficult problem, for which no commonly accepted solution exists – or at least we are not aware of one. A statistical comparison of the Gaussian Mixture Model (GMM) against alternative density models representing a continuous structure could be useful to refute the hypothesis of functional cell types if good candidates for alternative models existed and such models yielded a higher likelihood. However, such comparisons would never be strong evidence in favor of discrete clusters, because one would have to test against all alternative models. In addition, it is not clear to us what a good alternative model would be.
>
> Thus, the question will necessarily have to be answered qualitatively to some extent, and – as you also acknowledge – verified by experiments. As these additional experiments are technically very challenging, they first require a very clear hypothesis, which is what the present paper provides. The experimental verification, however, is clearly beyond the scope of a conference paper.
>
> Having said that, we followed your suggestion of applying our method to the readouts with randomly shuffled features (Figure A1 in the updated manuscript). The resulting t-SNE plot looks substantially less clustered than the one in Figure 6 and the MEIs within clusters also look less consistent. We believe this analysis is another piece of evidence for the functional cell types.
>
> We also investigated whether there is spurious structure in the clustering due to alignment procedure. As you suggested, we analyzed a synthetic example (Fig. B1 in the updated manuscript). We show examples of raw and aligned data as well as the t-SNE embeddings colored according to the GMM clustering. As the amount of noise increases, the t-SNE plots for the raw and the aligned data become increasingly similar. That suggests that despite some the overfitting to noise (as shown in Fig. 4), the alignment procedure does not significantly affect the GMM clustering or the t-SNE embeddings of the unstructured data.
>
> We would be very interested in any additional analyses we could do to convince you that we provide evidence for the hypothesis that the neurons form discrete clusters.
>
>
> [ML method is too specific]
>
> We agree that the presented alignment method is specific to the analysis of rotation-equivariant feature spaces. However, we think it might still be interesting to the community as a tool for analysis of equivariant feature spaces, since the same approach can be adapted to other symmetry groups (not only rotations) by replacing the cyclic shifts with an appropriate transformation for the other symmetry groups. There are multiple submissions to this conference (e.g. [1], [2], [3]) discussing equivariant (to rotations and other symmetries) neural networks, so we believe that additional tools for the analysis of such networks will be useful for future work in this direction.
>
> [1] https://openreview.net/forum?id=r1g6ogrtDr
> [2] https://openreview.net/forum?id=B1xtd1HtPS
> [3] https://openreview.net/forum?id=HJeYSxHFDS

---

> > ### Comment · AnonReviewer3 · 2019-11-11
> > **Thank you for careful rebuttal - additional requests**
> >
> > Thank you for the careful rebuttal and interesting complementary analyses.  In light of these additional results and arguments, I am ready to reconsider my evaluation if the following conditions are met:
> >
> > 1) removal of all ambiguous wording suggesting that functional cell types were identified in V1. In particular, the title and abstract sentence reproduced below are misleading:
> > (title) "Rotation-invariant clustering of functional cell types in primary visual cortex"
> > for example: cell types => cell responses
> > (abstract) "We [...] provide evidence that discrete functional cell types may exist in V1."
> > => unnecessarily ambiguous claim
> >
> > 2) Additional control: Similarly to A1, do a 2D t-SNE embedding of the aligned readouts R ̃ with feature weights randomly permuted. However, instead of permuting feature weights for each of the neurons, permute feature weights *across* neurons, so as to keep the marginal feature pooling statistics the same.

---

> > > ### Author Response · Authors · 2019-11-11
> > > **Title change and additional control**
> > >
> > > Thank you very much for a quick reply and additional suggestions.
> > >
> > > In the latest revision of the manuscript we:
> > > - Replaced “cell types” with “neuronal responses” in the title;
> > > - Updated the abstract to remove claims about the existence of cell types;
> > > - Following your suggestion, added Fig. A2 showing the t-SNE embedding of aligned readouts with the features randomly permuted across the neurons. Similarly to Fig. A1, the plot is substantially less clustered than the one in Fig. 6. The examples of MEIs within clusters also look less consistent in comparison to Fig. 6. In Fig. A3 we show the plot from Fig. 6 as well the two controls side by side for an easier comparison.

---

> > > > ### Comment · AnonReviewer3 · 2019-11-11
> > > > **Changing recommendation to Accept**
> > > >
> > > > Thank you for the swift reply. With these interesting additional controls and the softened claims, I now think that this article is a valuable contribution that should be published at ICLR.

---

### Official Review · AnonReviewer1 · 2019-10-30
**Official Blind Review #1876**

**Rating:** 8

**Review:**

The paper proposes an original approach to predict the function of groups of neurons in the V1 cortex based on their invariance to well designed rotation invariant CNN filters. The design of these features is funded by the observation that specific ganglion cell types have rotation and scale invariant responses to visual stimuli.
The method is very clearly explained and the evaluation on an publicly available dataset looks promising. The clustering Figure 6 in particular is very insightful.
The paper could have been more impactful if a comparison with a ground truth was built. The issue is clearly that ground truth is hard to establish for this type of problems but biological observations and annotations of cell types can be available (unfortunately not public as far as I know).
I would also be curious to know how such a method can be applied to a blind patient whose retina does not react to visual stimuli. Is there a biological function that will still preserve such invariance properties which allow to find structure in the data?

**Experience Assessment:**

I have published one or two papers in this area.

**Review Assessment: Checking Correctness Of Derivations And Theory:**

I assessed the sensibility of the derivations and theory.

**Review Assessment: Checking Correctness Of Experiments:**

I assessed the sensibility of the experiments.

**Review Assessment: Thoroughness In Paper Reading:**

I read the paper thoroughly.

---

> ### Author Response · Authors · 2019-11-09
> **A reply to Reviewer 1**
>
> Thank you for the positive assessment of our submission.
>
> We agree that a comparison to ground truth would be great. Unfortunately for pyramidal cells (> 80% of cells in cortex) it is unknown whether they’re further subdivided. There is evidence for genetic differences, but as far as we know there is no combined functional + genetic data available for pyramidal cells – certainly not to us.
>
> Regarding your question what would happen if there was no input from the retina, we could only speculate. We do not think there is reason to believe that there is a biological mechanism enforcing equivariance. It’s more likely to be a consequence of the statistics of the visual input. Also note that the fact that a rotation-equivariant representation works well to describe the data does not mean that the brain’s representation is actually equivariant.

---

### Decision · Program_Chairs · 2019-12-19

**Decision:**

Accept (Talk)

**Comment:**

This paper is enthusiastically supported by all three reviewers. Thus an accept is recommended.